# Whispering LLaMA: A Cross-Modal Generative Error Correction Framework for Speech Recognition

**Srijith Radhakrishnan**[1,2,6], **Chao-Han Huck Yang**[1,3,4], **Sumeer Ahmad Khan**[1,6], **Rohit Kumar**[1], **Narsis A. Kiani**[5], **David Gomez-Cabrero**[1], **Jesper N. Tegner**[1,5]

[1]King Abdullah University of Science and Technology   [2]Manipal Institute of Technology
[3]Georgia Institute of Technology   [4]NVIDIA Research   [5]Karolinska Institute
[6]SDAIA-KAUST Center of Excellence in Data Science and Artificial Intelligence
`srijithrkr@gmail.com; huckiyang@gatech.edu`

## Abstract

We introduce a new cross-modal fusion technique designed for generative error correction in automatic speech recognition (ASR). Our methodology leverages both acoustic information and external linguistic representations to generate accurate speech transcription contexts. This marks a step towards a fresh paradigm in generative error correction within the realm of n-best hypotheses. Unlike the existing ranking-based rescoring methods, our approach adeptly uses distinct initialization techniques and parameter-efficient algorithms to boost ASR performance derived from pre-trained speech and text models. Through evaluation across diverse ASR datasets, we assess our fusion technique, demonstrating a 37.66% improvement in word error rate (WER) relative performance compared to the n-best Oracle. To encourage future research, we have made our code and pre-trained models open source at `https://github.com/Srijith-rkr/Whispering-LLaMA`.

## 1 Introduction

End-to-end (E2E) trained speech models have demonstrated state-of-the-art performance on Automatic speech recognition (ASR) tasks. Several methods (Xia et al., 2017; Guo et al., 2019; Hu et al., 2021b; Yang et al., 2021a; Salazar et al., 2020) have widely adopted a two-pass rescoring paradigm to leverage upon language models to further enhance the capabilities of these models. In the two-pass paradigm, the first pass ASR system "generates" n-best hypotheses using an E2E acoustic model, while the second pass "re-ranks" these hypotheses by incorporating a language model (LM).

This two-pass reranking approach has several notable advantages over single-pass End-to-End (E2E) ASR systems (Amodei et al., 2016; Chan et al., 2016). Firstly, the subsequent large language model often captures a more comprehensive understanding (Stooke et al., 2023; Tur and De Mori,

2011) of language structures beyond the knowledge of transcribed audio present in the ASR model's pre-training data, thereby improving performance on unseen words. Furthermore, adapting the two-pass paradigm to accommodate domain shifts (Li et al., 2023; Liu et al., 2021; Yu et al., 2023) is much easier as only the language model needs to be fine-tuned on the new dataset. This alleviates the need for a spoken transcription corpus, which can be particularly beneficial for under-resourced or endangered spoken languages.

The recent emergence of conversational abilities in large language models, such as ChatGPT (OpenAI, 2023a) and GPT-4 (OpenAI, 2023b), has further sparked interest in leveraging the representational power of large pre-trained models for more complex tasks involving diverse data modalities (Yang et al., 2021b; Chang et al., 2023). Moreover, this new research direction also introduces a set of unique challenges related to considering information from other input modalities, such as acoustic and visual conditions (Peng et al., 2023; Zhang et al., 2023), in which could enrich using context beyond text-only input.

Recognizing speech signals is a task that necessitates both acoustic information (Hu et al., 2021a; Hung et al., 2023) (e.g., speaking environments) and linguistic information (Meng et al., 2023; Chen et al., 2023b,c) (e.g., context and domains). Efficiently amalgamating or integrating representation learning from acoustic modeling into language modelling to bolster its performance represents a notably intricate research domain that warrants further exploration. In this paper, we present a token-level fusion framework, merging two foundation (large-scale pre-trained) models into a recognition error correction paradigm, with the objective of enhancing the performance of ASR systems.

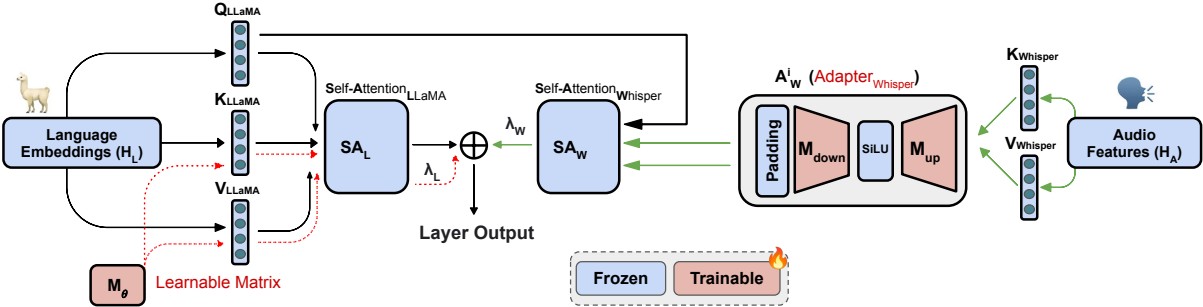

Figure 1: Illustration of proposed generative ASR error correction with a trainable token ($M_\theta$) and fusion mechanism inside a self-attention layer described in Section 3.2. A detailed model-wise illustration is discussed in Fig 2.

## 2 Related Work on ASR Post-processing

Transformer-based language models (Shin et al., 2019; Salazar et al., 2020) approach the two-pass paradigm by utilizing the summation of negative log-likelihoods of individual tokens from the language model to re-score the n-best output. Recent works on deliberation method (Hu et al., 2020; Prabhavalkar et al., 2018) and audio-attention based rescoring (Futami et al., 2021; Gandhe and Rastrow, 2020; Tanaka et al., 2021) in improving ASR-LM rescoring with the incorporation of acoustic features. Recent works on decoder prompting (Yang et al., 2023a) and encoder-decoder based error correction (Chen et al., 2023a; Ma et al., 2023) have demonstrated benefits in using an external language model for reducing the transcription error rate. Meanwhile, how to inject or fuse representations from a large acoustic model into another language model remains under investigation.

## 3 Method

We discuss the model architecture and the intuition behind the proposed feature combination in Section 3.1. The cross-modal fusion mechanism and weight initialization are explained in Section 3.2 and Section 3.3, respectively.

### 3.1 Generative Error Correction for ASR

Our approach combines two pre-trained models, Whisper (Radford et al., 2022) and LLaMA (Touvron et al., 2023), to facilitate generative error correction (Yang et al., 2023a; Chen et al., 2023a). Firstly, we employ Whisper, a multi-task encoder-decoder-based transformer (Vaswani et al., 2017) speech model trained on 680,000 hours of multilingual data, to encode audio representations and generate transcripts of n-best hypotheses. Secondly, we utilize LLaMA, a decoder-based large

language transformer model, to generate error-corrected transcripts by utilizing the n-best hypotheses via prompt (illustrated in Appendix, Fig 5) and audio representations via our proposed framework as input.

Whisper utilizes the encoder of a Transformer model to derive features from audio input, which are then fed into the decoder through multi-headed cross-attention, enabling auto-regressive text token prediction (Wang et al., 2023; Irie et al., 2022). The encoded features provide information from audio input via cross-attention, while the decoder's self-attention attends previous tokens using a key-value caching mechanism.

We fuse the audio features and the Whisper linear layers that generate the key and value pairs in the decoder's cross-attention mechanism to the LLaMA model to inject audio information. The inherent self-attention modules in LLaMA combined with the added cross-attention module make it analogous to the Whisper decoder. An overview of the proposed method is presented in Appendix, Fig. 2.

### 3.2 Cross-Modal Fusion Mechanism

We introduce our mechanism in Fig 1. To efficient fine-tune large models, we incorporate two residual adapter (Houlsby et al., 2019; Radhakrishnan et al., 2023; Chen et al., 2023d; Yang et al., 2023b) modules ($A_L^i$ and $A_W^i$) after the self-attention modules ($SA_L^i$) of the frozen LLaMA model at each layer. The first variable $A_L^i$ represents the adapter in layer $i$ used to fine-tune the LLaMA model using a scaled dot product attention mechanism. The second variable $A_W^i$ refers to another adapter in layer $i$ used to fuse Whisper features with the LLaMA model by following an autoencoder mechanism.

In each $A_L^i$, we incorporate a learnable matrix

$M_\theta^i \in \mathbb{R}^{\mathbf{N}_\theta \times \mathbf{N}_L}$. $\mathbf{N}_\theta$ denotes the dimensionality of the adapter embeddings, while $\mathbf{N}_L$ indicates the dimensionality of LLaMA embeddings. The language embedding feature extracted from the pre-trained LLM is represented by $H_L^i$ for each layer.

We repurpose the frozen LLaMA linear layers $K_{llama}^i$ and $L_{llama}^i$ from the LLaMA self-attention $\text{SA}_L^i$ to transform $M_\theta^i$ into key and value pairs, thus reducing the number of trainable parameters. We also reuse the query tensor from the frozen LLaMA self-attention module $\text{SA}_L^i$ to compute $A_L^i$, as shown below; $S$ represents the Softmax:

$$S\left(\frac{Q_{llama}^i(H_L^i) \cdot K_{llama}^i\left(M_\theta^i\right)^T}{\sqrt{d_k}}\right) V_{llama}^i\left(M_\theta^i\right) \tag{1}$$

To integrate the audio representations and key-value tensors from the Whisper decoder cross-attention module into the LLaMA model, we introduce two additional linear frozen transformations ($K_{whisper}^i$ and $V_{whisper}^i$) at each layer of the LLaMA model. These transformations are initialized with the respective weights from the cross-attention module of the Whisper decoder. By applying the audio representations to these additional linear transformations, we generate the key-value pairs that mirror the ones produced by Whisper. We then utilize the second adapter module $A_W^i$, to add trainable components to learn cross-modal representation. We apply a learnable projection matrix $M_{down}^i \in \mathbb{R}^{\mathbf{N}_W \times \frac{\mathbf{N}_W}{r}}$ to down project the obtained key and value pairs. Where $\mathbf{N}_W$ denotes the size of the Whisper encoded audio representations ($x$). We then apply the SiLU activation function (Elfwing et al., 2018) followed by a learnable up-projection $M_{up}^i \in \mathbb{R}^{\frac{\mathbf{N}_W}{r} \times \mathbf{N}_W}$, to compute trainable output:

$$A_W^i(x) \leftarrow \text{SiLU}\left(x \cdot M_{down}^i\right) M_{up}^i. \tag{2}$$

Using this setup, we transform the key-value pair at each layer to merge the hidden representation ($H_A$) from the output of the Whisper frozen pre-trained encoder with decoder from LLaMA:

$$\hat{K}_{whisper}^i \leftarrow A_W^i(K_{whisper}^i(H_A)); \tag{3}$$

$$\hat{V}_{whisper}^i \leftarrow A_W^i\left(V_{whisper}^i(H_A)\right). \tag{4}$$

Once we obtain the corresponding Whisper key and value pairs, we apply the padding mechanism described in 3.3 to preserve the latent structure of the Whisper Key and Value embeddings and

Table 1: Dataset sample statistics are provided with alias names. The Science & Technology category of GigaSpeech (Chen et al., 2021) is divided into two subsets: $\text{GS}_{SS}$ (small) and $\text{GS}_{SM}$ (medium), to evaluate performance differences with respect to data size.

| Dataset | Train | Test |
|---|---|---|
| ATIS (Hemphill et al., 1990) | 4978 | 893 |
| GigaSpeech: Entertainment ($\text{GS}_E$) | 4709 | 1000 |
| People & Blogs ($\text{GS}_P$) | 6802 | 1000 |
| Science & Technology ($\text{GS}_{SS}$) | 6908 | 1000 |
| Science & Technology ($\text{GS}_{SM}$) | 10323 | 1000 |

adjust the shape of $\hat{K}_{whisper}^i$ and $\hat{V}_{whisper}^i$ to enable computation of Multi Head Attention (MHA) with $Q_{llama}^i(H_L)$ from the frozen LLaMA model as before to obtain its adaptable self-attention head ($\text{SA}_W^i$) as:

$$S\left(\frac{Q_{llama}^i(H_L^i) \cdot \left(\hat{K}_{whisper}^i\right)^T}{\sqrt{d_k}}\right) \hat{V}_{whisper}^i \tag{5}$$

Then, we utilize a gated fusion mechanism, $\mathcal{W}$hispering-$\mathcal{L}$LaMA ($\mathcal{WL}$), to fuse all the modules together as shown below:

$$\text{SA}_{\mathcal{WL}}^i \leftarrow \text{SA}_L^i + \lambda_L \cdot A_L^i + \lambda_W \cdot \text{SA}_W^i, \tag{6}$$

where $\lambda_L$ and $\lambda_W$ are learnable scalars.

### 3.3 Weight Initialization

The latent dimensions of the Whisper and LLaMA models are different, making it necessary to reshape the Whisper tensors to match the shape of the LLaMA model while preserving the latent structure and information inherent in the Whisper model. Tensors are shaped in the format of $[B, NH, T, HS]$, which denotes the Batch size, Number of heads, context length and Head Size, respectively. The last two dimensions undergo transformation during the attention mechanism. Hence in order to preserve the Whisper latent structure, We initialize a matrix of zeros of shape $\in \mathbb{R}^{NH_{llama} \times T_{whisper} \times HS_{llama}}$ and fill the principal diagonal of the last two dimensions with ones. We then place $\hat{K}^i$ and $\hat{V}^i$ on the top left corner of the padding template. We further initialize the projection matrices $M_{down}^i, M_{up}^i$ on the second adapter module $A_W^i$ as identity matrices. The proposed framework encounters significant losses and fails to converge unless this initialization strategy is followed to preserve Whisper's latent representations.

Table 2: The experimental results are presented in terms of WER without text normalization. The performance of our proposed framework is reported in rows $2 - 4$. Oracle refers to the candidate among the n-best hypothesis with the lowest word error rate compared to the ground truth. Rows $5 - 9$ represent different ablation experiments on the best-performing model, $\mathcal{WL}_M$. The WERR is measured relative to the oracle performance as shown in B.2

| # | Method | #Para. | ATIS | $GS_E$ | $GS_P$ | $GS_{SS}$ | $GS_{SM}$ | $WER_{Avg}(\downarrow)$ | WERR ($\uparrow$) |
|---|--------|--------|------|--------|--------|-----------|-----------|------------------------|-------------------|
| 1 | Oracle (1st-pass) | - | 13.76 | 28.22 | 22.84 | 23.93 | 19.5 | 21.64 | - |
| 2 | $\mathcal{WL}_L$ | 26.40M | 2.04 | 21.76 | 19.21 | 20.55 | 11.6 | 15.03 | 30.52 |
| 3 | $\mathcal{WL}_M$ | 7.97M | 1.77 | **21.61** | **16.20** | **18.02** | **9.82** | **13.48** | **37.66** |
| 4 | $\mathcal{WL}_S$ | 4.89M | 1.89 | 22.24 | 17.23 | 19.157 | 10.185 | 14.144 | 34.62 |
| 5 | $\mathcal{WL}_M$ w/o masking | 4.89M | 3.94 | 27.56 | 18.10 | 21.71 | 12.79 | 20.04 | 22.25 |
| 6 | $\mathcal{WL}_M$ w/o $H_A$ | 4.89M | 253.20 | 123.19 | 203.44 | 376.81 | 256.44 | 242.61 | -1020.68 |
| 7 | $\mathcal{WL}_M$ w/o init. | 4.89M | 405.83 | 500.58 | 414.34 | 461.63 | 390.64 | 434.60 | -1907.45 |
| 8 | $\mathcal{WL}_M$ w/o $SA_W$ | 1.22M | 1.66 | 24.99 | 18.734 | 20.73 | 10.86 | 15.39 | 28.83 |
| 9 | Big-scale Adapter | 4.91M | **1.45** | 23.65 | 16.59 | 19.93 | 10.62 | 14.45 | 33.21 |

## 4 Experimental Setup

### 4.1 Models

For our experiments, we utilize the LLaMA-7B model architecture. As we instruct the language model with the generated hypotheses (as explained in Section 4.3.1) to perform generative error correction, we initialize our model weights with Alpaca (Taori et al., 2023), a model fine-tuned from LLaMA-7B, utilizing 52,000 instruction-following demonstrations to enable instruction following abilities. To extract audio representations from input audio clips, we employ Whisper-Large V2, a model with 1.55B parameters trained on 620,000 hours of audio data. Additionally, we employ Whisper-Tiny, a model with 70M parameters, for generating our transcripts, as described in the subsequent section 4.2. We name our model Whispering LLaMA ($\mathcal{WL}$) and train three variants with our proposed framework with $\mathbf{N}_\theta = 10$ and $r = 8, 16, 32$ named $\mathcal{WL}_L$ (large), $\mathcal{WL}_M$ (medium), $\mathcal{WL}_S$ (small), respectively. We design $\mathcal{WL}_L$ with two separate $A_W$ adapter modules for key and value, respectively. $\mathcal{WL}_M$ and $\mathcal{WL}_S$ use the same $A_W$ adapter in section 3.2 to reduce trainable parameters.

### 4.2 Dataset

We curate our own transcripts by leveraging two datasets: the Airline Travel Information System (Hemphill et al., 1990) (ATIS) and GigaSpeech (Chen et al., 2021). ATIS consists of audio recordings of individuals querying flight information. GigaSpeech, contains audio from audiobooks, podcasts and YouTube videos on diverse topics. ATIS represents a semantically correct, domain-specific dataset, while GigaSpeech represents a more noisy, real-world setting in our eval-

uation. We select domain-specific subsets in GigaSpeech and focus on three specific categories: Entertainment, People and Blogs, and Science and Technology. To explore performance variations with respect to the number of data points, we further divide the Science and Technology category into two subsets. Table 1 provides detailed information on the number of training points per dataset. We chose Whisper-Tiny to generate the n-best hypothesis baseline to establish a robust evaluation environment that aligns more closely with real-world settings dealing with sub-optimal hypotheses. By employing Whisper-Tiny, we mimic a *weak* acoustic model with lower-quality hypotheses. Feeding LMs with better-quality hypotheses from Whisper-Large would make the generative error correction task less challenging for LM adaptation and does not explore the model's performance under practical settings where our method is intended to be employed. However, we emphasize that our method remains effective when starting with a Whisper-Large hypothesis in Appendix E.

For each audio clip, we generate 200 hypotheses using a top-k value of 200 and a randomly selected temperature between the range of $[0.7, 0.8]$. Subsequently, we filter out redundant sentences and select the top 15 with the highest log probability.

### 4.3 Training Pipeline

The input to our model consists of the encoded audio representations extracted from the Whisper-Large model, accompanied by the 15-best transcripts generated by Whisper-Tiny. We employ the prompt template used by the Alpaca model as shown in Appendix Fig 5. We utilize the Adam optimizer (Kingma and Ba, 2014) and experiment with learning rates of $1 \times 10^{-2}$, $1 \times 10^{-3}$, and

$5 \times 10^{-4}$, selecting the optimal value. The model is trained for 25 epochs, employing early stopping to prevent overfitting. Training is conducted on two Nvidia A100 GPUs to leverage efficient parallel processing. An effective batch size of 32 is used, and a weight decay of $1 \times 10^{-2}$ is applied.

### 4.3.1 LLM Prompting Examples for ASR

We employ the Alpaca (Taori et al., 2023) prompt template, as illustrated in Fig. 5 of the Appendix, to generate the n-best hypotheses. This template features an instructional segment designated by the **Instruction** tag, which offers guidance to the model. Essential contextual data required by the model is housed under the Input tag. The prompt concludes with the **Response** tag, directing the model to enact the specified instruction within the supplied input context. Rather than adopting the recent advances of Task-Activating Prompting (Yang et al., 2023a) (TAP), we opt to feed the LLM with its task-specific data (e.g., speech recognition in our instance). Our alternative approach facilitates second-pass error correction, mitigating the latency issues observed in the extensive context windows of the TAP-based generative ASR error correction.

### 4.4 Performance Studies

Results from our experiments have been reported in Table 2. The $\mathcal{WL}_M$ model achieves the best performance with a word-error-rate relative (WERR) of 37.66%, as defined in B.2. A comparison between $\mathcal{WL}_L$ and $\mathcal{WL}_M$ indicates that having separate adapter modules for key and value pairs does NOT result in performance improvements. Further dataset-specific analyses are detailed in Appendix B. The models exhibit better performance on the Gigaspeech with more in-domain data.

### 4.5 Ablation Studies

We empirically discover that masking the prompt except for the ground truth in the cross entropy loss function significantly improves the performance. We attribute this improvement to the model's enhanced capacity to grasp accurate semantics, achieved by refraining from penalizing the model for erroneous sentences found in the n-best hypotheses. Row 5 represents the performance of $\mathcal{WL}_M$ without masking. We further investigate if the proposed framework is utilizing the audio representations from Whisper by substituting them with random tensors generated from a normal distribution as the input (Row 6). Additionally, we

explore the significance of our weight initialization mechanism by replacing it with random initialization (Row 7). Both of these ablation studies validate our intuition, demonstrating that the method utilizes acoustic features effectively and highlight the importance of the initialization mechanism in preserving the latent structure of the acoustic embeddings. For further insights, please refer to Appendix D. We also remove the Whisper adapter ($\text{SA}_W$) module for an text feature only baseline performance using adapters (Row 8). Since the disparity between the number of trainable parameters is high, we train another model with an increased adapter context dimension of $\mathbf{N}'_{\theta} = 4\mathbf{N}_{\theta}$ (Row 9).

## 5 Conclusion

We propose a novel framework to leverage the external knowledge from LLM to improve the transcription accuracy of ASR systems. Our framework presents a parameter-efficient way to integrate large foundational Speech and Language models to achieve competitive WERR improvements. We further conduct extensive ablation experiments to validate our intuitions and open source our code and pretrained-weights to the research community.

## 6 Limitation

Using large models such as LLaMA is intuitive, as it provides a comprehensive comprehension of language structure owing to its internet-scaled pretraining. However, deploying these systems and conducting research with them in real-world scenarios is challenging due to their computationally intensive nature. In our approach, we aim to design our framework to be parameter-efficient by **re-using** multiple model components with adapters for model fusion. Nonetheless, incorporating audio representations into the training pipeline extends the training duration by 394.76%. This underscores the significance of alignment issues (Yen et al., 2023). Furthermore, our proposed solution demonstrates a need for a larger volume of data to achieve optimal performance despite having a modest parameter count of only 7.97M to integrate foundational models. During our experimentation, we encountered issues related to over-fitting on datasets. To mitigate this problem, we trained with a reduced learning rate and monitored the Word Error Rate (WER) performance throughout the training process and selected the model checkpoint with the best performance to implement early stopping.

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

## A  Appendix

In this Appendix, We investigate the performance difference between datasets in Section B, and provide illustrations of the model-level architectural design in Section C. Section D provides more insight into the results from ablation studies and we report a Whisper Large Hypothesis baseline in Section E.

## B  Dataset Analysis

We report the WER from our experiments before and after text normalization on Table 3. We convert the model prediction and the ground truth to lower-case and remove punctuation during text normalization. The ATIS dataset is not impacted by text normalization because the dataset does not contain any punctuation. It only contains contractions such as *"I'd like"* instead of *"I would like"*. ATIS consists of audio recordings of individuals querying automated airline travel inquiry systems for flight information. We believe the lack of punctuation and the consistent structure present within the ATIS dataset enables improved WER performance compared to GigaSpeech. The Gigaspeech dataset contains punctuation and lacks consistency within the dataset because it has diverse categories and sources such as audiobooks, podcasts and YouTube videos.

### B.1  More Discussion on Ground Truth Match Rate

During dataset generation, we remove the ground truth if it is present among the Whisper generated n-best hypotheses. This allows us to introduce a new metric called Ground Truth Match Rate (GTMR). GTMR calculates the percentage of predictions generated by the model that exactly match the ground truth. This metric indicates the model's ability to learn the structure of the dataset. The GTMR of our experiments before and after text normalization is reported in Table 5. The model is able to learn the structure of the dataset better with more data points, as observed from the performance difference between $\mathbf{GS}_{SS}$ and $\mathbf{GS}_{SM}$. It can also be observed that the model is able to learn the simpler structure of ATIS much better than other GigaSpecch datasets.

### B.2  WERR

Word error rate relative is calculated as

$$WERR(i) \leftarrow \frac{Oracle(i) - WER(i)}{Oracel(i)} \times 100 \quad (7)$$

where $Oracle(i)$ refers to the average Oracle performance in terms of WER and $WER(i)$ refers to the average performance of a particular method.

## C  Proposed Architecture Illustrations

We present a model-level overview of our proposed method described in Section 3.2 in Fig 2. We add two modules into each layer of the LLaMA model. The LLaMA adapter and the Fusion adapter which refer to $A_L$ and $A_W$, respectively. We initialize the Fusion adapter with the weights from the Whisper cross-attention module in the decoder model. LLaMA takes the encoded features generated by the Whisper encoder and the n-best hypothesis generated by the Whisper in a prompt format as input to generate the error-corrected response.

## D  Failure Case Studies of Generative ASR with Whispering-LLaMA

Since the WER error rate in row 6 ( $\mathcal{WL}_M$ w/o audio representations ) and row 7 ( $\mathcal{WL}_M$ w/o initialization) of table 2 is high and provides no insight into model performance, we present the training loss graphs of the best performing model ( $\mathcal{WL}_M$) with and without audio representations in Figure 4. The model is not able to converge below a certain threshold without audio representations. Additionally, we include the training loss graphs of $\mathcal{WL}_M$ with and without our initialization mechanism in Figure 3. Without our initialization mechanism, the latent structure of the Whisper encoder embedding is not preserved, leading to an inability to converge.

## E  Whisper Large Decoding Baseline

We report the results of using the hypothesis generated by Whisper Large to train our best-performing model ( $\mathcal{WL}_M$) on GigaSpeech Entertainment ($GS_E$) and Science and Technology ($GS_{SS}$) datasets on Table 4. By leveraging the LLaMA model with the proposed generative error correction mechanism, we are able to match the performance of the Whisper Large model with 1.5 billion parameters by using a Whisper-Tiny model with just 70 million parameters. Using the hypotheses generated by Whisper Large results in a higher

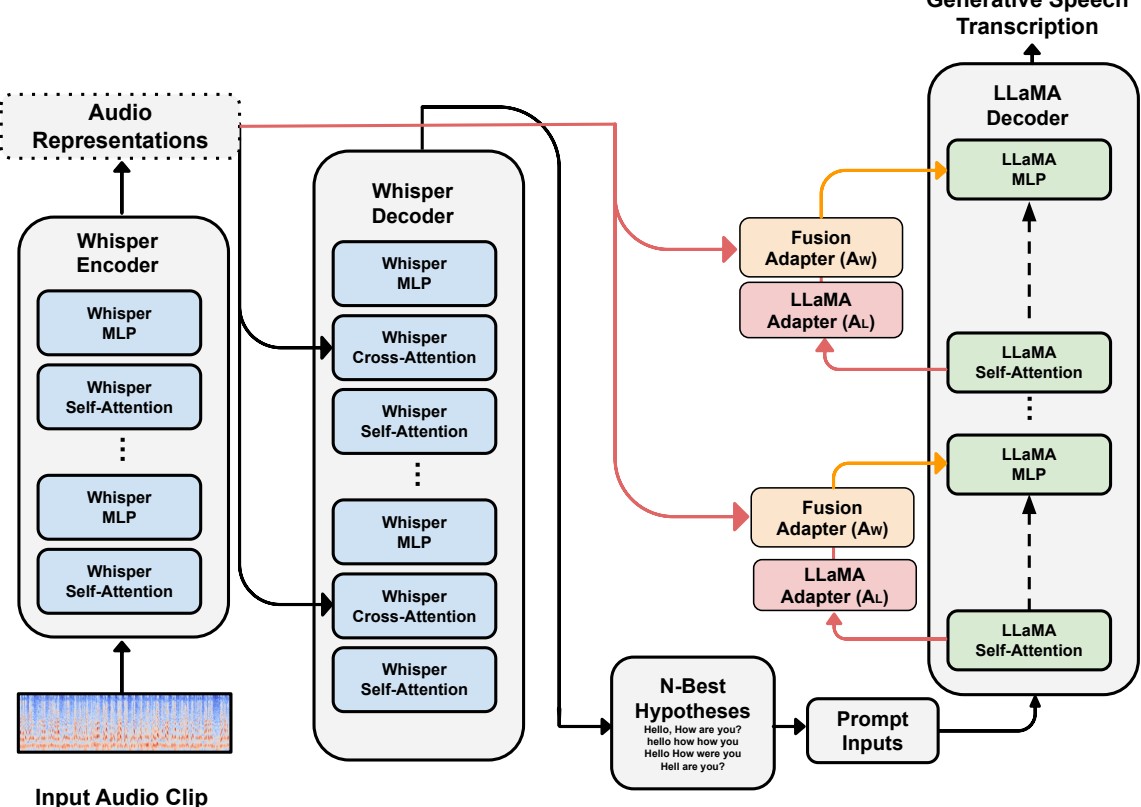

Figure 2: Whispering-LLaMA model-overview of proposed adaptation pipeline described in Section 3.2

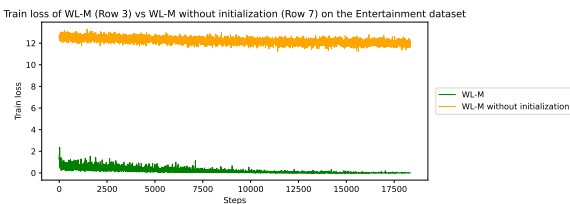

Figure 3: Train loss of $\mathcal{WL}_M$ (Row 3) vs $\mathcal{WL}_M$ without initialization (Row 7) on the Entertainment dataset

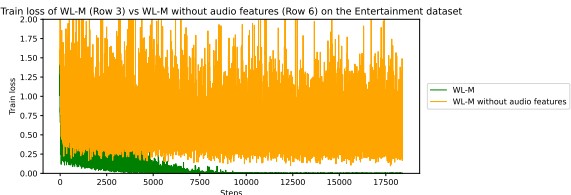

Figure 4: Train loss of $\mathcal{WL}_M$ (Row 3) vs $\mathcal{WL}_M$ without audio representations (Row 6) on the Entertainment dataset

WERR as expected. This finding confirms the effectiveness of the proposed approach, particularly in the context of Whisper Large generated N-best hypotheses.

Table 3: The experimental results in terms of Word Error Rate (WER), before and after text normalization. We convert all text to lowercase and remove the following punctuation [".", "-", "?", "'"]. Rows 1-3 represent before text normalization, and Rows 4-6 represent after text normalization.

| # | Method | ATIS | $\mathbf{GS}_E$ | $\mathbf{GS}_P$ | $\mathbf{GS}_{SS}$ | $\mathbf{GS}_{SM}$ | $\mathbf{WER_{Avg}}(\downarrow)$ |
|---|--------|------|------|------|------|------|------|
| 1 | $\mathcal{WL}_L$ | 2.04 | 21.76 | 19.21 | 20.55 | 11.6 | 15.03 |
| 2 | $\mathcal{WL}_M$ | 1.77 | 21.61 | 16.20 | 18.02 | 9.82 | 13.48 |
| 3 | $\mathcal{WL}_S$ | 2.11 | 23.60 | 17.13 | 20.16 | 10.73 | 14.75 |
| 4 | $\mathcal{WL}_L$ | 2.04 | 14.71 | 13.36 | 14.23 | 7.44 | 10.35 |
| 5 | $\mathcal{WL}_M$ | 1.77 | 17.71 | 10.83 | 12.00 | 6.07 | 9.67 |
| 6 | $\mathcal{WL}_S$ | 1.89 | 15.21 | 11.49 | 13.03 | 6.41 | 9.61 |

## F  Reproducibility Resources

We have open-sourced the pre-trained model weights and code, available at `https://github.com/Srijith-rkr/Whispering-LLaMA`. Our future plan includes integrating this baseline into both Espnet (Watanabe et al., 2018) and HyPoradise (Yang et al., 2023a; Chen et al., 2023a) to accommodate a broader range of use cases.

### Acknowledgement

The authors thank Maxim Lvov and anonymous reviewers for their feedback on the draft.

**Model Input**

### Instruction:

You are an ASR transcript selector. You have a few transcripts generated by an automatic speech recognition model. Your task is to generate the most likely transcript from them. If the generated transcripts have grammatical or logical errors, you will modify them accordingly to produce the most accurate and coherent transcript.

### Input:
so that it can carry its momentum to the logic
that he can carry his moment on tour with the logic
that he can carry his momentum through with the logic
that he can carry his momentum to with an logic
that he can carry his momentum true within logic
that he carries momentum through with logic
that it can carry a moment and through with the logic
that it can carry a moment of truth with the logic
that it can carry a moment on tour with the logic
that it can carry a moment on true with the logic
that it can carry a momentum tool within logic
that it can carry as a moment on tour with the logic
that it can carry at moments and through with the logic
that it can carry his moment on tour with a logic
that it can carry his moment on tour with the logic

### Response:

**Model Output**

that it can carry its momentum through with a logic.

Figure 5: Illustration of the Alpaca prompt template used in our proposed framework

Table 4: Results of employing a Whisper Large generated hypothesise baseline in comparison to the proposed Whisper Tiny hypothesis baseline

| Method | $\text{GS}_E$ | $\text{GS}_{SS}$ | $\textbf{WER}_{\textbf{Avg}}(\downarrow)$ | $\textbf{WERR}(\uparrow)$ |
|---|---|---|---|---|
| Oracle with Whisper Tiny | 28.22 | 23.93 | 26.08 | - |
| $\mathcal{WL}_M$ with Whisper Tiny | 21.61 | 18.02 | 19.82 | **24.01** |
| Oracle with Whisper Large | 21.78 | 18.09 | 19.93 | - |
| $\mathcal{WL}_M$ with Whisper Large | 15.59 | 12.77 | 14.18 | **28.86** |

Table 5: The experimental results in terms of Ground Truth Match Rate (GTMR), Before and after text normalization. Rows 1-3 represent before text normalization, and Rows 5-7 represent after text normalization. Row 4 and 6 denote the average GTMR across datasets, and column $WER_{Avg} \downarrow$ denotes the average GTMR across each method

| # | Method | ATIS | $\text{GS}_E$ | $\text{GS}_P$ | $\text{GS}_{SS}$ | $\text{GS}_{SM}$ | $\textbf{WER}_{\textbf{Avg}}(\downarrow)$ |
|---|---|---|---|---|---|---|---|
| 1 | $\mathcal{WL}_L$ | 86.1 | 26.5 | 24.1 | 24.8 | 36.2 | 39.5 |
| 2 | $\mathcal{WL}_M$ | 88.3 | 26.3 | 31.4 | 28.8 | 41.0 | 43.17 |
| 3 | $\mathcal{WL}_S$ | 87.5 | 25.2 | 27.6 | 26.6 | 38.6.1 | 41.10 |
| 4 | Avg | 87.3 | 26.0 | 27.7 | 26.7 | 38.6 | - |
| 5 | $\mathcal{WL}_L$ | 86.1 | 46.9 | 46 | 45.5 | 56.4 | 56.2 |
| 6 | $\mathcal{WL}_M$ | 88.3 | 47.0 | 54.3 | 49.2 | 62.7 | 60.3 |
| 7 | $\mathcal{WL}_S$ | 87.5 | 44.3 | 49.5 | 46.3 | 60.0 | 57.5 |
| 8 | Avg | 87.3 | 46.0 | 49.9 | 47.0 | 59.7 | - |