# OpenReview forum: "Whispering LLaMA: A Cross-Modal Generative Error Correction Framework for Speech Recognition"
_EMNLP/2023/Conference — EMNLP 2023 Main_

### Official Review · Reviewer_7p1W · 2023-08-01

**Soundness:** 4

**Excitement:**

4: Strong: This paper deepens the understanding of some phenomenon or lowers the barriers to an existing research direction.

**Paper Topic And Main Contributions:**

This paper introduces a cross-modal technique for correction in ASR, which combines both acoustic data and linguistic data. This is a new paradigm to inject acoustic features from a large acoustic model to a large language model. Experiments verify the effectiveness of this method.

**Questions For The Authors:**

1. Why is adapter A_L^i introduced? It does not fuse any acoustic information.

2. Did you try the LoRA adapter? It's a more widely used adapter for fine-tuning LLMs.



**Reasons To Accept:**

1. This paper proposed a novel and effective fusion mechanism for incorporating acoustic features into LLMs.

2. The results are promising and the ablation study is detailed.

**Reasons To Reject:**

1. The oracle n-best hypo baseline is a smaller model, which reduces the validity of performance comparison.

**Reproducibility:**

4: Could mostly reproduce the results, but there may be some variation because of sample variance or minor variations in their interpretation of the protocol or method.

**Reviewer Confidence:**

3: Pretty sure, but there's a chance I missed something. Although I have a good feel for this area in general, I did not carefully check the paper's details, e.g., the math, experimental design, or novelty.

---

> ### Author Rebuttal · Authors · 2023-08-27
>
> The authors wish to express their heartfelt appreciation for awarding a soundness score of 4 and acknowledging the originality of our work. We have provided a related discussion to address the reviewer's questions below.
> ***
> #### **Regarding using Whisper Tiny for n-best hypothesis**
> We chose the Whisper Tiny generated oracle baseline due to the following reasons:
>
> **Uniform training and testing evaluation:** We use the Whisper Tiny model to generate the N-best oracle baseline as we trained our models with the hypothesis generated by the Whisper Tiny model.
>
> **Better Evaluation Criteria:** The proposed method tries to leverage the rich pretrained knowledge in Large Language Models as text normalisation to improve the transcription accuracy of audio models. In this paradigm, providing the Language model with better-quality hypothesis from Whisper Large makes the generative error correction task less challenging for the language model, and does not explore the model's performance under practical settings where our method is intended to be employed. As real-world use cases may encompass scenarios where the hypotheses from acoustic models might exhibit lower quality.
>
> **Research Reproducibility:** We adopted the Whisper Tiny model to ensure research reproducibility. Generating multiple hypotheses using the Whisper Large model proves computationally intensive for most research environments. To illustrate, Generating hypothesis for the $GS_{S}$ dataset (11,323 data points) takes 1 hour 48 minutes by the Whisper Tiny model in a single NVIDIA A100-80GB GPU, while it takes around 37 hours with Whisper Large.
>
> As per the reviewer’s interest, we also agree that having a model trained on the Whisper Large hypothesis would be a great assessment to the community and would enhance the comprehensibility of the paper. We have attached the results of using the hypothesis generated by Whisper Large for training our best-performing model (WL-M) on GigaSpeech Entertainment ($GS_{E}$) and Science & Technology ($GS_{S}$) datasets to investigate the performance. We could not run the experiments across all the datasets due to limited time. However, **_We commit to adding a related discussion_** to the camera-ready submission that depicts the performance when using the Whisper Large model to generate hypotheses across all datasets.
>
>
> | |$GS_{E}$|$GS_{S}$|$WER_{Avg}\downarrow$|$WERR\uparrow$|
> |:----|:----|:----|:----|:----|
> |Oracle with Whisper Tiny (Proposed method) |28.22|23.93|26.08|- |
> |WL-M with Whisper Tiny (Proposed method) |21.61|18.02|19.82|**24.01**|
> |Oracle with Whisper Large|21.78|18.09|19.93|-|
> |WL-M with Whisper Large|15.59|12.77|14.18|**28.86**|
>
> Our results showcase the effectiveness of our proposed method in benefiting both the Whisper Large and Whisper Tiny hypothesis-trained models. We sincerely hope that these findings offer valuable insights into the validity of performance comparison.
>
> ***
> #### **Q1: Why is Adapter $A_L^i$ introduced?**
>
> The proposed method is centred around the utilisation of two adapter modules: the $A_L$ adapter aims to fine-tune the LLaMA model, while the $A_W$ adapter focuses on the integration of acoustic features. Using only the $A_W$ adapter would apply transformations to audio features from the Whisper model, but it would not introduce any trainable components inside the language model to use the input audio features. Consequently, the necessity for the $A_L$ Adapter remains, as we still need trainable parameters inside the language model injected by $A_L$ to fine-tune the LLaMA Model.
> ***
> #### **Q2: Did you try the LoRA adapter?**
>
> We greatly appreciate your recommendation regarding the LoRA adapter. We did not incorporate it during the time of this submission. However, through subsequent experimentation in our Lab, we found that the Adapter approach still performs better than LoRA. This could be because LoRA has no nonlinearity to map complex boundaries. We have attached some results (WER) from our LoRA implementation below for reference.
>
> |                                           | *$GS_{E}$*  | *$GS_{S}$*  | *$GS_{M}$* |
> | ----------------------------------------------- | ------ | ------ | ----- |
> | WL_M (Proposed method) | 21.61  | 18.02 | 9.82 |
> |                 Replacing Adapters with LoRA of rank 16                                | 23.38 | 20.36 | 10.09 |

---

### Official Review · Reviewer_YmwX · 2023-08-03

**Soundness:** 2

**Excitement:**

3: Ambivalent: It has merits (e.g., it reports state-of-the-art results, the idea is nice), but there are key weaknesses (e.g., it describes incremental work), and it can significantly benefit from another round of revision. However, I won't object to accepting it if my co-reviewers champion it.

**Paper Topic And Main Contributions:**

The paper approaches the problem of correcting speech recognition errors with the use of of LLMs.
The authors propose to utilize cross-attention tensors from Whisper (a SOTA ASR model) as a part of the LLM decoder (in this case LLaMA) and then use the resulting adapted LLM for speech error correction.

**Questions For The Authors:**

1. What is the purpose of using Alpaca in place of a pure LLaMA?

2. Why the whisper-tiny results are corrected instead of whisper-large results?

**Reasons To Accept:**

The paper addresses the issue of ASR-LLM integration which is a problem of practical important that raised to the significance in the recent years.

The method seams to be promising. However, it should be evaluated in a more demanding setting.

**Reasons To Reject:**

1. The paper uses whisper-large to improve the results of whisper-tiny. I expected that the authors would show the improvement to WER and WERR of whisper-large with the use of their method. Evaluating on whisper-tiny leads to overly optimistic results. Whisper-tiny is just a "toy" model to be used in tutorials and examples. In any real application one of the larger Whisper models would be used.

2. The performance analysis in Section 4.4 is not compatible with the results reported in  tables. In particular:

a) line 272: 32.83 WERR is not reported in Table 2.
b) line 273: There is no WL model mentioned in any of the tables.

**Reproducibility:**

4: Could mostly reproduce the results, but there may be some variation because of sample variance or minor variations in their interpretation of the protocol or method.

**Reviewer Confidence:**

4: Quite sure. I tried to check the important points carefully. It's unlikely, though conceivable, that I missed something that should affect my ratings.

**Typos Grammar Style And Presentation Improvements:**

line 225: "with of our"

---

> ### Author Rebuttal · Authors · 2023-08-28
>
> The authors sincerely thank the reviewer for their dedicated effort in reviewing the submission. Their attention to detail contributed significantly to enhancing the manuscript's quality and provided performance-driven insight that benefited from the review cycle.
>
> We have provided a related discussion to address their questions below.
>
> ***
> #### **Addressing and clarifying discrepancies in the manuscript.**
> a) In response to your observation on line 272 regarding the reported WERR value of 32.83%, we acknowledge this discrepancy. The accurate value should be 37.66%, reflecting the performance improvement of the $WL-M$ model from Table 2.
>
> b) Your observation on line 273 pertaining to the absence of a $WL$ model reference in our tables is correct. The intended reference should have been $WL-M$ rather than $WL$, reflecting the model under discussion.
>
> We deeply appreciate your attention to detail. We apologise for this error and assure you that we will rectify these mistakes in our final version.
>
> ***
> #### **Q1: Purpose of using Alpaca in place of a pure LLaMA?**
>
> The LLaMA-**1** [1] model open-sourced by Meta has **no instructional fine-tuning** [2] and, hence, is not aligned towards following instructions. In our proposed framework, we **instruct** the language model with the generated hypothesis  (as illustrated in Figure 5) to perform generative error correction. Hence, we use Alpaca [3],  A version of the LLaMA model that has been fine-tuned on instructional datasets, as it can adhere to provided instructions better than LLaMA.
>
> ***
> #### **Q2: Why the whisper-tiny results are corrected instead of whisper-large results?**
>
> We chose the Whisper Tiny generated hypothesis due to the following reasons:
>
> **Better Evaluation Criteria:** The proposed method tries to leverage the rich pretrained knowledge in Large Language Models as text normalisation to improve the transcription accuracy of audio models. In this paradigm, providing the Language model with better-quality hypothesis makes the generative error correction task less challenging for the language model, and does not explore the model's performance under practical settings where our method is intended to be employed. As real-world use cases may encompass scenarios where the hypotheses from acoustic models might exhibit lower quality.
>
> **Research Reproducibility:** We adopted the Whisper Tiny model to ensure research reproducibility. Generating multiple hypotheses using the Whisper Large model proves computationally intensive for most research environments. To illustrate, Generating hypothesis for the $GS_{S}$ dataset (11,323 data points) takes 1 hour 48 minutes by the Whisper Tiny model in a single NVIDIA A100-80GB GPU, while it takes around 37 hours with Whisper Large.
>
>
>
> As per the reviewer’s interest, we also agree that having a model trained on the Whisper Large hypothesis would be a great assessment to the community and would enhance the comprehensibility of the paper. We have attached the results of using the hypothesis generated by Whisper Large for training our best-performing model (WL-M) on GigaSpeech Entertainment ($GS_{E}$) and Science & Technology ($GS_{S}$) datasets to investigate the performance. We could not run the experiments across all the datasets due to limited time. However, **_We commit to adding a related discussion_** to the camera-ready submission that depicts the performance when using the Whisper Large model to generate hypotheses across all datasets.
>
>
> | |$GS_{E}$|$GS_{S}$|$WER_{Avg}\downarrow$|$WERR\uparrow$|
> |:----|:----|:----|:----|:----|
> |Oracle with Whisper Tiny (Proposed method) |28.22|23.93|26.08|- |
> |WL-M with Whisper Tiny (Proposed method) |21.61|18.02|19.82|**24.01**|
> |Oracle with Whisper Large|21.78|18.09|19.93|-|
> |WL-M with Whisper Large|15.59|12.77|14.18|**28.86**|
>
> Our results showcase the effectiveness of our proposed method in benefiting both the Whisper Large and Whisper Tiny hypothesis-trained models. Using the hypothesis generated by Whisper Large results in a higher WERR as expected. We sincerely hope that these findings will convincingly showcase the advantages of our approach to the reviewer.
>
> ***
> **References**
>
> [1]: Hugo Touvron and Thibaut Lavril and Gautier Izacard and Xavier Martinet and Marie-Anne Lachaux and Timothée Lacroix and Baptiste Rozière and Naman Goyal and Eric Hambro and Faisal Azhar and Aurelien Rodriguez and Armand Joulin and Edouard Grave and Guillaume Lample. 2023. LLaMA: Open and Efficient Foundation Language Models
>
> [2]: Chung, H.W., Hou, L., Longpre, S., Zoph, B., Tay, Y., Fedus, W., Li, E., Wang, X., Dehghani, M., Brahma, S. and Webson, A., 2022. Scaling instruction-finetuned language models. arXiv preprint arXiv:2210.11416.
>
> [3]: ''tatsu-lab/stanford_alpaca" repository in GitHub

---

### Official Review · Reviewer_v94L · 2023-08-04

**Soundness:** 3

**Excitement:**

3: Ambivalent: It has merits (e.g., it reports state-of-the-art results, the idea is nice), but there are key weaknesses (e.g., it describes incremental work), and it can significantly benefit from another round of revision. However, I won't object to accepting it if my co-reviewers champion it.

**Paper Topic And Main Contributions:**

The paper describes an attempt to combine the Whisper ASR encoder with the LLama LLM. In contrast to conventional methods like n-best reranking, the two models are fused by means of "cross-modal fusion", i.e. the whisper's audio features are integrated into LLama's frozen self-attention module using trainable adapter modules followed by a weighted fusion. Experimental results indicate significant reductions in WER compared to an oracle n-best selection for ATIS and GigaSpeech test sets.

**Reasons To Accept:**

The described approach is new and interesting and should be discussed at the conference.

**Reasons To Reject:**

It appears that it will be very difficult to reproduce the results - the authors point out themselves that a lot of fiddling has to be done with initialization, monitoring WER on the test set and stopping the training at the right moment.

**Reproducibility:**

2: Would be hard pressed to reproduce the results. The contribution depends on data that are simply not available outside the author's institution or consortium; not enough details are provided.

**Reviewer Confidence:**

3: Pretty sure, but there's a chance I missed something. Although I have a good feel for this area in general, I did not carefully check the paper's details, e.g., the math, experimental design, or novelty.

---

> ### Author Rebuttal · Authors · 2023-08-24
>
> The authors appreciate the reviewer's effort in reviewing the submission and are deeply grateful for recommending the paper to be discussed at the conference. We have provided a related discussion to address their concern regarding the reproducibility of our proposed framework.
>
> **_In the interest of reproducibility, we will open source the model code, training script, inference script and pre-trained weights under the MIT license._**
>
> ***
> #### **Regarding initialisation mechanism**
> The initialisation mechanism in our proposed framework ensures that the latent structure and information inherent in the audio features are preserved during injection to the LLaMA model and **are implemented as part of the training script** yet to be open-sourced (currently included in supplementary materials). The results in Table 1 (rows 2-4) and the ablation studies in Row 7 (WL-M w/o init.) consistently showcase the stability and effectiveness of the proposed initialization mechanism.
>
> We further conducted 3 runs with our best-performing model $WL-M$ on the GigaSpeech Entertainment ($GS_{E}$), People & Blogs ($GS_{P}$) and Science & Technology ($GS_{S}$) datasets to investigate the consistency in results and have reported the WER below.
>
> |     | Run 1  | Run 2  | Run 3  | Mean   | Std   |
> | --- | ------ | ------ | ------ | ------ | ----- |
> | $GS_{E}$ | 21.496 | 21.529 | 21.518 | 21.514 | **0.017** |
> | $GS_{M}$ | 18.01  | 17.969 | 17.958 | 17.979 | **0.027**|
> | $GS_{S}$ | 16.498 | 16.282 | 16.255 | 16.345 | **0.133** |
> ***
> #### **Regarding Early stopping**
>
> Early stopping is an established countermeasure against overfitting with Adapters, as demonstrated in prior work [1], and is easily reproducible. We also **commit to integrating the early stopping feature** into training scripts when we open-source our work.
>
> We sincerely hope that these measures address any concerns surrounding the reproducibility of our work.
>
> ***
> **References**
>
>
> [1] : Thuy-Trang Vu, Shahram Khadivi, Dinh Phung, and Gholamreza Haffari. 2022. Domain Generalisation of NMT: Fusing Adapters with Leave-One-Domain-Out Training. In Findings of the Association for Computational Linguistics: ACL 2022, pages 582–588, Dublin, Ireland. Association for Computational Linguistics.

---

### Meta-Review · Area_Chair_ueHx · 2023-09-19

**Recommendation:** 4

**Metareview:**

**Originality:**

This work is the logical next step for integrating LLMs with ASR models. Many similar prior works have explored this space, albeit only very recently. This work differs from others, for instance (Ma et al. 2023), primarily in the scale of the model  (Llama) and data (Gigaspeech) used for fine-tuning the error-correction model, as well as in the type of LLM (decoder only). The more novel aspect of this work is the fact that it gives LLMs more direct access to acoustic representations, through the Whisper encoder, rather than just through n-best lists or ASR outputs, through cross-attention.

**Significance:**

This work, as well as the topic of leveraging LLMs for ASR error correction is very significant.

**Clarity:**

The paper is well-written though there are a number of minor grammatical errors that could be fixed. As pointed out by some of the reviewers, some corrections to tables need to be made.

**Pros:**
   - A new method addressing an incredibly important problem
   - Thorough experimentation
   - Promising results

**Cons:**
   - This method should be evaluated in other settings.
   - Some minor concerns about reproducibility.

**General Comments**
Large pre-trained models such as Whisper and Llama are trained on large amounts of web-scraped corpora. Without knowledge of the training data, It is difficult to ascertain to what extent data leaking (training data in the test sets) is a problem. Evaluation sets must be carefully selected to ensure conclusions hold across data sets, and to limit the potential for data leakage. In this study most results are presented on web-scraped corpora. the results from ATIS are very encouraging, but it would be useful to see results on more datasets.

The authors seem to have addressed many of the concerns about using Whisper Tiny as well reproducibility.

---

### Decision · Program_Chairs · 2023-10-07

**Decision:**

Accept-Main

**Comment:**

**Originality:**

This work is the logical next step for integrating LLMs with ASR models. Many similar prior works have explored this space, albeit only very recently. This work differs from others, for instance (Ma et al. 2023), primarily in the scale of the model  (Llama) and data (Gigaspeech) used for fine-tuning the error-correction model, as well as in the type of LLM (decoder only). The more novel aspect of this work is the fact that it gives LLMs more direct access to acoustic representations, through the Whisper encoder, rather than just through n-best lists or ASR outputs, through cross-attention.

**Significance:**

This work, as well as the topic of leveraging LLMs for ASR error correction is very significant.

**Clarity:**

The paper is well-written though there are a number of minor grammatical errors that could be fixed. As pointed out by some of the reviewers, some corrections to tables need to be made.

**Pros:**
   - A new method addressing an incredibly important problem
   - Thorough experimentation
   - Promising results

**Cons:**
   - This method should be evaluated in other settings.
   - Some minor concerns about reproducibility.

**General Comments**
Large pre-trained models such as Whisper and Llama are trained on large amounts of web-scraped corpora. Without knowledge of the training data, It is difficult to ascertain to what extent data leaking (training data in the test sets) is a problem. Evaluation sets must be carefully selected to ensure conclusions hold across data sets, and to limit the potential for data leakage. In this study most results are presented on web-scraped corpora. the results from ATIS are very encouraging, but it would be useful to see results on more datasets.

The authors seem to have addressed many of the concerns about using Whisper Tiny as well reproducibility.